

# Impact of examined lymph node count on long-term survival of T1-2N0M0 double primary NSCLC patients after surgery: a SEER study

Kan Jiang[1,*], Xiaohui Zhi[1,*], Yue Shen[2], Yuanyuan Ma[1], Xinyu Su[1] and Liqing Zhou[1]

[1] Department of Radiation Oncology, the Affiliated Huai'an Hospital of Xuzhou Medical University, the Second People's Hospital of Huai'an, Huai'an, Jiangsu, China
[2] Shandong Provincial Key Laboratory of Radiation Oncology, Cancer Research Center, Shandong Cancer Hospital Affiliated to Shandong University, Shandong Academy of Medical Sciences, Jinan, Shandong, China
* These authors contributed equally to this work.

## ABSTRACT

**Purpose:** The relationship between examined lymph nodes (ELN) and survival has been confirmed in several single early-stage malignancies. We studied the association between the ELN count and the long-term survival of T1-2N0M0 double primary non-small cell lung cancer (DP-NSCLC) patients after surgery, based on the Surveillance, Epidemiology and End Results (SEER) database.

**Methods:** A total of 948 patients were identified and their independent prognostic factors were analyzed. These factors included the ELN count, which related to overall survival (OS) and the cancer-specific survival (CSS) of synchronous ($n$ = 426) and metachronous ($n$ = 522) T1-2N0M0 DP-NSCLC patients after surgery.

**Results:** X-tile analysis indicated that the cutoff value for the sum of ELNs was 22 for both OS and CSS in the synchronous DP-NSCLC group. Patients with a sum of ELNs >22 were statistically more likely to survive than those with ≤22 ELNs. X-tile analysis revealed that the ELN count of the second lesion was related to both OS and CSS in the metachronous DP-NSCLC group. The optimal cutoff value was nine. These results were confirmed using univariate and multivariate Cox regression analyses.

**Conclusion:** Our findings indicate that ELN count was highly correlated with the long-term survival of T1-2N0M0 double primary NSCLC patients after surgery.

## INTRODUCTION

Lung cancer is the leading cause of cancer-related mortality worldwide (*Bray et al., 2018*). More advanced imaging technology has improved the diagnostic rate of multiple primary lung cancer (MPLC) (*Thakur et al., 2018*). Double primary lung cancer (DPLC) which defined as two primary malignances developed at the same or different time made up the

Corresponding authors
Xinyu Su, 15261798461@163.com
Liqing Zhou, zlq-hill@163.com

majority of MPLC (*Guo et al., 2017*). Early-stage lung cancers reportedly account for most cases of DPLC (*Liu et al., 2002*). Surgery is the primary treatment for early-stage lung cancer when lesions are limited to T1-2N0M0, according to the National Comprehensive Cancer Network guidelines. The complete resection of the lesions, or a lobectomy with lymph node examination, improves the survival rate of early-stage node-negative MPLC, with a 5 year survival rate of over 50% (*Vansteenkiste et al., 2014*).

The relationship between examined lymph nodes (ELN) and survival was confirmed in several studies on early-stage malignancies; a higher ELN count is correlated with a better prognosis. *Liu et al. (2019)* found it prognostically significant when more than 12 ELNs were present in pT1N0M0 esophageal cancer. *Ji et al. (2017)* studied gastric cancer and found that more than 22 ELNs would improve overall survival. Similar results were found in early-stage single primary non-small cell lung cancer (NSCLC) (*Ou & Zell, 2008*; *Varlotto et al., 2009*). However, there is no current research on the association between ELN and multiple primary cancers.

Therefore, we conducted a population-based, retrospective investigation of the Surveillance, Epidemiology and End Results (SEER) database to explore the relationship between ELN count and the long-term survival of T1-2N0M0 double primary NSCLC (DP-NSCLC) patients after surgery.

## MATERIALS AND METHODS

### Data source
The data in this study were extracted from the SEER program of the National Cancer Institute, accounting for approximately 28% of the total US population from 18 regions. All cases were derived from the SEER 18 Regs Research Data + Hurricane Katrina Impacted Louisiana Cases, November 2015 Sub (1973–2013), using SEER*Stat 8.3.5 software.

### Study population
We identified a total of 10,552 patients who had been diagnosed with DPLC between 1988–2013. The inclusion criteria were as follows: (1) patients were pathologically confirmed NSCLC according to "ICD-O-3 Hist/behave"; (2) the first and second cancers were pathologically different and proved to be primary cancer according to "Primary by international rules"; (3) patients with T1-2N0M0 stage of both primary cancers were screened and restaged based on the American Joint Committee on Cancer's 8th edition TNM stage according to the original stage and tumor size offered; (4) both primary lesions received lobectomy; (5) active follow-up status of more than 1 month; (6) definite information existed for the ELN; and (7) ELN were confirmed by pathology. The exclusion criteria were as follows: (1) patients with missing or unknown clinical information (race, primary site, grade); and (2) patients who underwent any type of radiation or chemotherapy. A total of 948 patients were included in this study after these criteria were met (Fig. 1).

Patients were divided into a synchronous double primary NSCLC (SDP-NSCLC) group ($n = 426$) and a metachronous double primary NSCLC (MDP-NSCLC) group ($n = 522$).

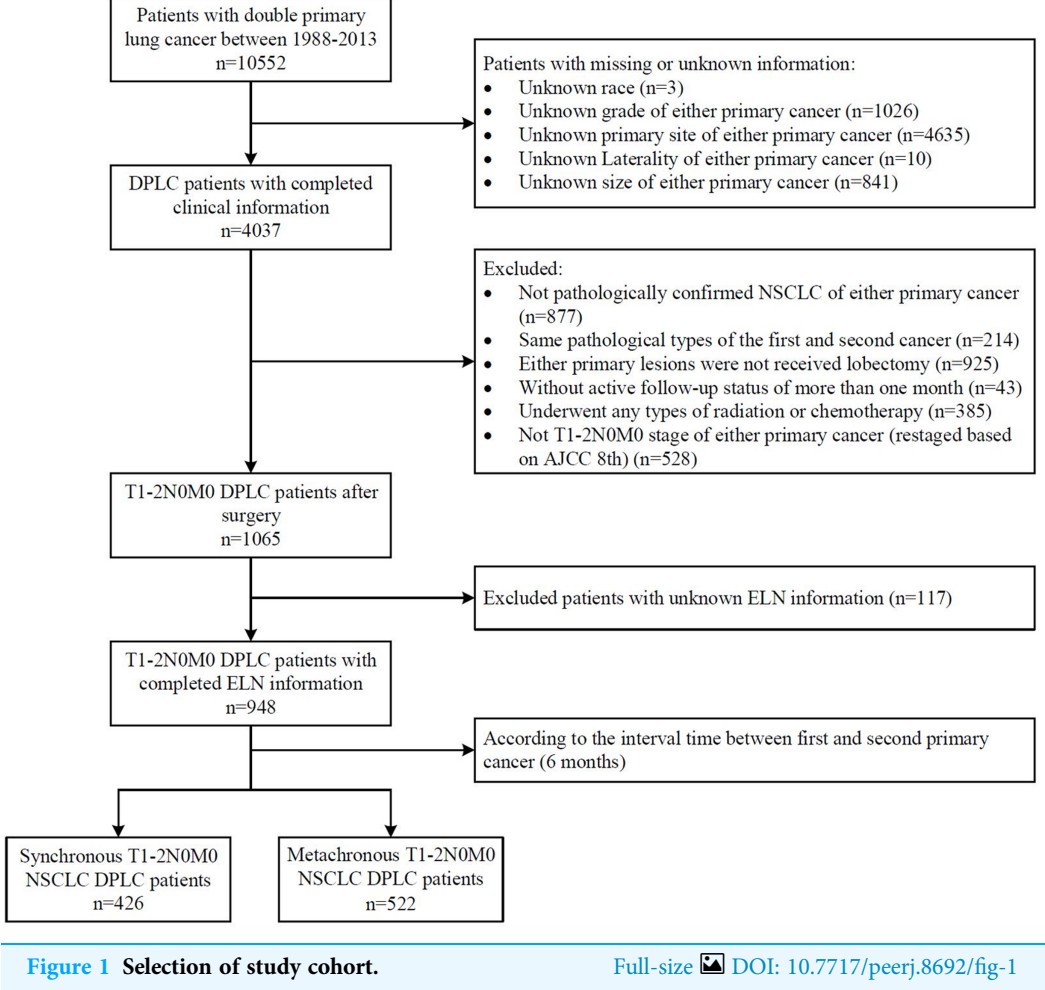

**Figure 1 Selection of study cohort.**

SDP-NSCLC group patients were those with second primary NSCLC identified within 6 months after the first primary NSCLC was diagnosed. MDP-NSCLC was defined as the second primary NSCLC detected more than 6 months after the diagnosis of first primary NSCLC. The following variables were collected for each patient: demographics (age at diagnosis of the first primary NSCLC, sex, race), the clinicopathological characteristics of both the first and second primary NSCLC (primary site, laterality, grade, tumor size, ELN), and follow-up data (survival months, survival status, cancer-specific death). Two primary lesions were occasionally discovered concurrently in patients in the SDP-NSCLC group, making it impossible to distinguish the first primary cancer. In these cases, the lesions were taken into consideration along with the patient to compare the clinicopathological characteristics between the two groups. Prognostic factors were analyzed in the SDP-NSCLC group using the following variables: (1) grade max (the worse grade of the two lesions); (2) size max (the larger size of the two lesions); (3) ELN plus (the sum of the ELN of the two lesions). The ELN count was determined by the SEER code, "Regional nodes examined", which recorded the total number of regional lymph nodes

removed and examined by the pathologist. All of the uncertain ELN data was deleted (code 90, 96, 97, 98, 99).

OS and cancer-specific survival (CSS) were set as the endpoints of our study. OS was calculated as the period from the date of the first primary NSCLC diagnosis to the date of death from any cause. CSS was defined as the period from the initial diagnosis until death due to NSCLC. The interval time refers to the period between the detection of the first primary NSCLC and the development of the second primary NSCLC in the MDP-NSCLC group. This study was approved by the Ethics Committee of the Affiliated Huai'an Hospital of Xuzhou Medical University.

### Statistical analysis

The chi-squared test or Fisher's exact test were applied for categorical variables and the two independent samples $t$-test was applied to continuous variables in order to assess the differences between the baseline characteristics of the SDP-NSCLC and MDP-NSCLC groups. The OS of patients in each group was estimated using the Kaplan–Meier method and applied log-rank tests were used to compare the survival curves. X-tile software version 3.6.1 (Yale University, New Haven, CT, USA) was used to determine the cutoff value of the number of ELN. X-tile can be used to assess the relationship between a biomarker and patient outcome and to discover the cut-points based on marker expression. Univariate Cox proportional hazards models were applied to all variables for prognostic analysis to identify the predictive factors associated with OS and CSS in each group. Multivariate analysis was used on predictive factors ($P \leq 0.1$) to determine the independent prognostic factors. A two-sided $P < 0.05$ was considered to be statistically significant. We performed all analyses using SPSS statistics version 23.0 (IBM Corporation, Armonk, NY, USA).

## RESULTS

### Patient characteristics

A total of 948 DP-NSCLC patients who were diagnosed from 1988 to 2013 with resected stage T1-2N0M0 NSCLC, including 426 SDP-NSCLC and 522 MDP-NSCLC patients, were identified for this study. Table 1 summarizes the baseline patient characteristics of the two groups.

The median ages were 69 (range 32–95) and 67 (range 39–85), differed significantly ($P < 0.001$) between the SDP-NSCLC group and the MDP-NSCLC group, respectively. The MDP-NSCLC group was more likely to be bilateral, while the SDP-NSCLC group was similar on the ipsilateral and bilateral sides. No significant difference was noted in sex ($P = 0.906$) and race ($P = 0.047$) between the two groups. Statistical distinctions were noted in the primary site ($P = 0.003$) and grade ($P = 0.006$) between the SDP-NSCLC and MDP-NSCLC groups, with moderate differentiation (Grade II) and the greatest proportion of lesions located in the upper lobe. The median tumor size in the SDP-NSCLC group was 18 mm, while tumors in the MDP-NSCLC group measured 20 mm ($P = 0.005$). The median ELN count for all lesions was four, but differed significantly between two groups ($P = 0.018$).

**Table 1 Baseline clinical characteristics of patients and lesions in SDP-NSCLC group and MDP-NSCLC group.**

| Variable | SDP-NSCLC group $n$ (%) | MDP-NSCLC group $n$ (%) | $P$ |
|---|---|---|---|
| Patients (total) | 426 (100) | 522 (100) | |
| Age, median (range) | 69 (32–95) | 67 (39–85) | **<0.001** |
| Sex | | | 0.906 |
| Male | 182 (42.7) | 225 (43.1) | |
| Female | 244 (57.3) | 297 (56.9) | |
| Race | | | **0.047** |
| White | 367 (86.2) | 453 (86.8) | |
| Black | 44 (10.3) | 37 (7.1) | |
| Others | 15 (3.5) | 32 (6.1) | |
| Location | | | **<0.001** |
| Ipsilateral | 210 (49.3) | 129 (24.7) | |
| Bilateral | 216 (50.7) | 393 (75.3) | |
| Lesions (total) | 852 (100) | 1,044 (100) | |
| Primary site | | | **0.003** |
| Upper lobe | 530 (62.2) | 592 (56.7) | |
| Middle lobe | 46 (5.4) | 71 (6.8) | |
| Lower lobe | 276 (32.4) | 372 (35.6) | |
| Overlapping lesion | 0 (0.0) | 9 (0.9) | |
| Grade | | | **0.006** |
| I | 219 (25.7) | 207 (19.8) | |
| II | 360 (42.3) | 501 (48.0) | |
| III | 252 (29.6) | 319 (30.6) | |
| IV | 21 (2.4) | 17 (1.6) | |
| Laterality | | | 0.062 |
| Left | 348 (40.8) | 471 (45.1) | |
| Right | 504 (59.2) | 573 (54.9) | |
| Size (mm), median (range) | 18 (3–50) | 20 (3–50) | **0.005** |
| ELN, median (range) | 4 (0–44) | 4 (0–57) | **0.018** |

Notes:
SDP-NSCLC, synchronous double primary non-small cell lung cancer; MDP-NSCLC, metachronous double primary non-small cell lung cancer; ELN, examined lymph node.
The bold entries represent statistically significant.

## Number of ELNs and survival analysis

X-tile analysis in the SDP-NSCLC group showed that 0 and 22 were the first-rank cutoff values for ELN plus, in terms of both OS ($\chi^2$ = 18.271, $P$ < 0.001) and CSS ($\chi^2$ = 11.715, $P$ = 0.003) (Supplement Information 1). The optimal cutoff values for the ELN count of the second tumor for MDP-NSCLC patients were 0 and 9, which was statistically significance in OS ($\chi^2$ = 14.256, $P$ = 0.001) and CSS ($\chi^2$ = 17.051, $P$ < 0.001) (Supplement Information 2). However, the optimal cutoff values for the ELN count of the first tumor in the MDP-NSCLC group as determined by the X-tile analysis (2 and 8) were not statistically different for either OS ($\chi^2$ = 4.654, $P$ = 0.095) or CSS ($\chi^2$ = 4.604,

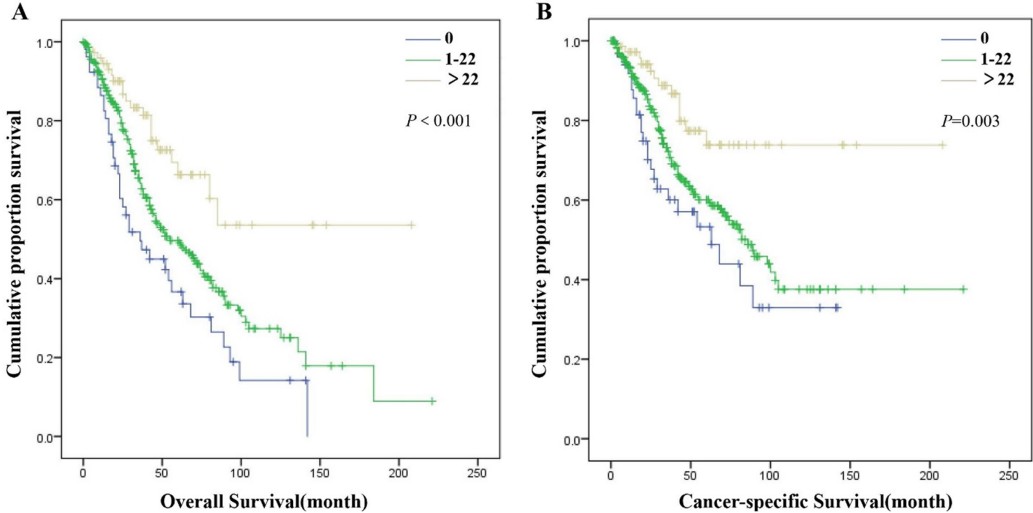

**Figure 2 Overall survival (A) and cancer-specific survival (B) of patients in SDP-NSCLC group with different ELN plus.**

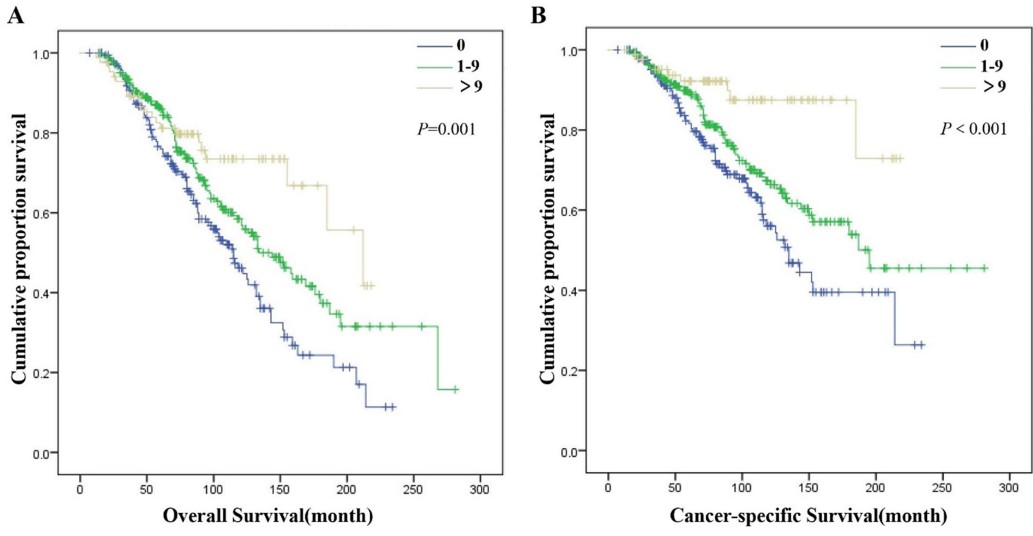

**Figure 3 Overall survival (A) and cancer-specific survival (B) of patients in MDP-NSCLC group with different ELN 2.**

$P = 0.100$). Univariate and multivariate Cox regression analyses were completed for further confirmation of the cutoff values of the ELN count. The greater number of ELN plus corresponded with improved OS (0 vs. 1–22: $P = 0.017$; 0 vs. >22: $P < 0.001$; 1–22 vs. >22: $P = 0.002$) and CSS for patients in the SDP-NSCLC group (0 vs. 1–22: $P = 0.046$; 0 vs. >22: $P < 0.001$; 1–22 vs. >22: $P = 0.007$) (Fig. 2; Supplement Information 3). In the MDP-NSCLC group, ELN 2 was the only independent prognostic factor of OS (0 vs. 1–9: $P = 0.009$; 0 vs. >9: $P < 0.001$; 1–9 vs. >9: $P = 0.043$) and CSS (ELN 2: 0 vs. 1–9: $P = 0.035$; 0 vs. >9: $P < 0.001$; 1–9 vs. >9: $P = 0.002$) (Fig. 3; Supplement Information 4).

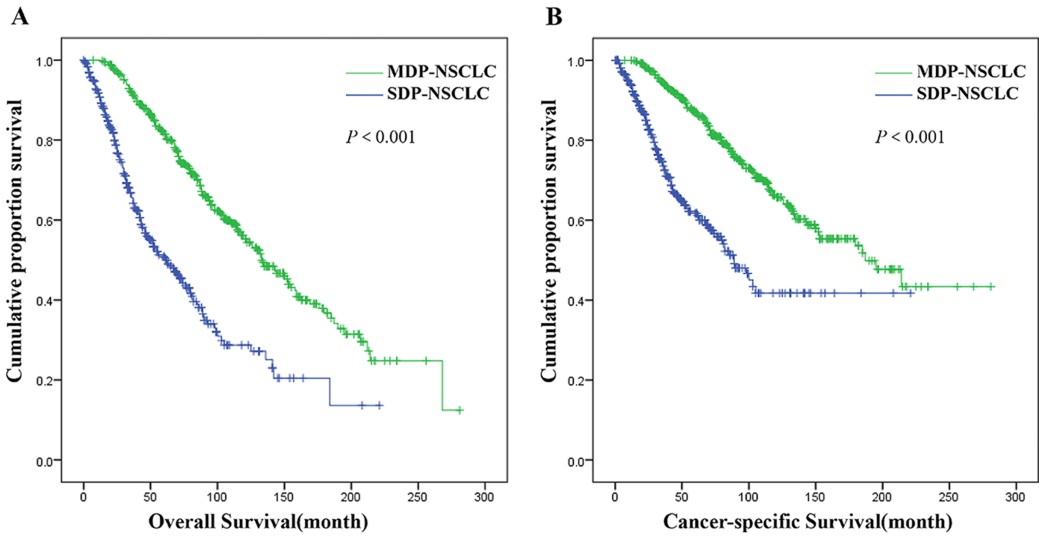

**Figure 4 Overall survival (A) and cancer-specific survival (B) of patients in SDP-NSCLC group vs. MDP-NSCLC group.**

## Cox proportional hazards regression model for OS and CSS

The median follow-up period was 93 months (range 1–281). A significant difference was observed in the OS and CSS for patients in the MDP-NSCLC group compared with the SDP-NSCLC group (Fig. 4), with a median OS of 133 months vs. 62 months, and a median CSS of 187 months vs. 89 months, respectively ($P < 0.001$).

The ELN count and other variables were analyzed using the univariate and multivariate Cox proportional hazards regression model to evaluate the prognostic factors of each group. The univariate and multivariate analyses for SDP-NSCLC and ELN plus indicated that age ($P = 0.001$) and sex ($P = 0.001$) were independent prognostic factors of OS, while age ($P = 0.027$), sex ($P = 0.006$) and grade max ($P = 0.022$) were independent prognostic factors of CSS (Table 2). With respect to the MDP-NSCLC group, apart from ELN2, age ($P < 0.001$), interval time ($P < 0.001$), primary site 2 ($P = 0.038$) and size 2 ($P < 0.001$) were significant after univariate and multivariate analyses of OS. The interval time ($P < 0.001$), grade 2 ($P = 0.048$), size 2 ($P = 0.002$) and ELN 2 were independent prognostic factors of CSS (Table 3).

## DISCUSSION

ELN count was shown to be highly correlated with the long-term survival of T1-2N0M0 double primary NSCLC patients after surgery, regardless of whether they were from the SDP-NSCLC group or MDP-NSCLC group. Regional lymph node examination performed during surgery was found to significantly prolong survival compared with lymph node conservation in both the SDP-NSCLC and MDP-NSCLC groups. In order to promote better survival, we recommend that a sum of more than 22 lymph nodes be examined during surgery for two lesions in SDP-NSCLC patients and more than nine lymph nodes be examined during surgery for the second lesion in MDP-NSCLC patients.

**Table 2 The prognostic factors associated with overall survival and cancer-specific survival of patients in SDP-NSCLC group by univariate and multivariate Cox regression.**

| Variable | Overall survival | | | | | | Cancer-specific survival | | | | | |
|---|---|---|---|---|---|---|---|---|---|---|---|---|
| | Univariate | | | Multivariate | | | Univariate | | | Multivariate | | |
| | HR | 95% CI | P | HR | 95% CI | P | HR | 95% CI | P | HR | 95% CI | P |
| Age | 1.029 | [1.013–1.045] | **<0.001** | 1.028 | [1.012–1.044] | **0.001** | 1.021 | [1.002–1.040] | **0.028** | 1.022 | [1.002–1.042] | **0.027** |
| Sex | | | | | | | | | | | | |
| Male | 1 | | | 1 | | | 1 | | | 1 | | |
| Female | 0.642 | [0.488–0.844] | **0.002** | 0.620 | [0.470–0.818] | **0.001** | 0.631 | [0.454–0.877] | **0.006** | 0.625 | [0.447–0.874] | **0.006** |
| Race | | | 0.215 | | | | | | 0.273 | | | |
| White | 1 | | | | | | 1 | | | | | |
| Black | 1.052 | [0.675–1.639] | 0.824 | | | | 1.003 | [0.587–1.714] | 0.992 | | | |
| Others | 0.416 | [0.154–1.125] | 0.084 | | | | 0.318 | [0.078–1.285] | 0.108 | | | |
| Primary site | | | 0.558 | | | | | | 0.336 | | | |
| Upper lobe | 1 | | | | | | 1 | | | | | |
| Middle lobe | 1.335 | [0.679–2.625] | 0.403 | | | | 1.078 | [0.437–2.656] | 0.871 | | | |
| Lower lobe | 1.128 | [0.839–1.515] | 0.425 | | | | 1.298 | [0.918–1.835] | 0.140 | | | |
| Second site | | | 0.176 | | | | | | 0.192 | | | |
| Upper lobe | 1 | | | | | | 1 | | | | | |
| Middle lobe | 1.298 | [0.769–2.191] | 0.329 | | | | 1.290 | [0.686–2.425] | 0.429 | | | |
| Lower lobe | 1.300 | [0.971–1.741] | 0.078 | | | | 1.370 | [0.967–1.941] | 0.077 | | | |
| Location | | | | | | | | | | | | |
| Ipsilateral | 1 | | | | | | 1 | | | | | |
| Bilateral | 0.924 | [0.702–1.215] | 0.570 | | | | 0.938 | [0.675–1.303] | 0.702 | | | |
| Grade max | | | 0.311 | | | | | | 0.053 | | | **0.022** |
| I | 1 | | | | | | 1 | | | 1 | | |
| II | 1.456 | [0.889–2.386] | 0.135 | | | | 2.208 | [1.127–4.324] | **0.021** | 2.483 | [1.261–4.888] | **0.009** |
| III | 1.546 | [0.956–2.501] | 0.076 | | | | 2.041 | [1.049–3.970] | **0.036** | 2.229 | [1.139–4.362] | **0.019** |
| IV | 1.752 | [0.850–3.612] | 0.129 | | | | 3.289 | [1.368–7.911] | **0.008** | 3.786 | [1.562–9.178] | **0.003** |
| Size max | 1.010 | [0.997–1.024] | 0.129 | | | | 1.013 | [0.997–1.029] | 0.122 | | | |
| ELN plus | | | **<0.001** | | | **<0.001** | | | 0.004 | | | 0.001 |
| 0 | 1 | | | 1 | | | 1 | | | 1 | | |
| 1–22 | 0.657 | [0.458–0.942] | **0.022** | 0.641 | [0.445–0.923] | **0.017** | 0.714 | [0.458–1.112] | 0.136 | 0.631 | [0.402–0.992] | **0.046** |
| >22 | 0.316 | [0.183–0.545] | **<0.001** | 0.304 | [0.176–0.527] | **<0.001** | 0.321 | [0.163–0.630] | **0.001** | 0.284 | [0.143–0.561] | **<0.001** |

**Notes:**
SDP-NSCLC, synchronous double primary non-small cell lung cancer; HR, hazard ratio; CI, confidence interval; ELN, examined lymph node.
The bold entries represent statistically significant.

We investigated the differences in clinical characteristics between the SDP-NSCLC and MDP-NSCLC groups and found that the patients with SDP-NSCLC were older than those with MDP-NSCLC, which is consistent with the research conducted by *Wang et al. (2019)*. Elderly patients are more at risk for being immunocompromised, which may account for the age difference in the populations. The lesions in our study were typically found the upper lobe of the lung, which is consistent with other studies (*Guo et al., 2017*). Double primary lesions frequently occurred in the ipsilateral or bilateral lung, similar

**Table 3 The prognostic factors associated with overall survival and cancer-specific survival of patients in MDP-NSCLC group by univariate and multivariate Cox regression.**

| Variable | Overall survival | | | | | | Cancer-specific survival | | | | | |
|---|---|---|---|---|---|---|---|---|---|---|---|---|
| | Univariate | | | Multivariate | | | Univariate | | | Multivariate | | |
| | HR | 95% CI | P | HR | 95% CI | P | HR | 95% CI | P | HR | 95% CI | P |
| Age | 1.058 | [1.039–1.076] | <0.001 | 1.035 | [1.016–1.055] | <0.001 | 1.045 | [1.024–1.067] | <0.001 | 1.019 | [0.996–1.043] | 0.099 |
| Sex | | | | | | | | | | | | |
| Male | 1 | | | 1 | | | 1 | | | | | |
| Female | 0.761 | [0.582–0.996] | **0.047** | 0.817 | [0.620–1.077] | 0.152 | 0.808 | [0.582–1.122] | 0.204 | | | |
| Race | | | 0.257 | | | | | | 0.431 | | | |
| White | 1 | | | | | | 1 | | | | | |
| Black | 0.827 | [0.480–1.423] | 0.492 | | | | 0.880 | [0.462–1.675] | 0.696 | | | |
| Others | 0.592 | [0.303–1.157] | 0.125 | | | | 0.591 | [0.260–1.341] | 0.208 | | | |
| Interval time | 0.977 | [0.972–0.982] | <0.001 | 0.976 | [0.971–0.981] | <0.001 | 0.975 | [0.969–0.981] | <0.001 | 0.974 | [0.968–0.981] | **<0.001** |
| Location | | | | | | | | | | | | |
| Ipsilateral | 1 | | | | | | 1 | | | | | |
| Bilateral | 1.033 | [0.756–1.412] | 0.837 | | | | 1.196 | [0.803–1.779] | 0.379 | | | |
| First tumor | | | | | | | | | | | | |
| Primaty site 1 | | | 0.339 | | | | | | 0.295 | | | |
| Upper lobe | 1 | | | | | | 1 | | | | | |
| Middle lobe | 1.483 | [0.913–2.411] | 0.112 | | | | 1.713 | [0.972–3.020] | 0.063 | | | |
| Lower lobe | 1.102 | [0.817–1.488] | 0.524 | | | | 1.116 | [0.774–1.608] | 0.558 | | | |
| Overlapping lesion | 0.453 | [0.063–3.241] | 0.430 | | | | 0.688 | [0.096–4.944] | 0.710 | | | |
| Grade I | | | 0.126 | | | | | | 0.198 | | | |
| I | 1 | | | | | | 1 | | | | | |
| II | 1.305 | [0.860–1.980] | 0.211 | | | | 1.358 | [0.809–2.280] | 0.247 | | | |
| III | 1.460 | [0.945–2.255] | 0.088 | | | | 1.607 | [0.940–2.747] | 0.083 | | | |
| IV | 2.629 | [1.087–6.359] | 0.032 | | | | 2.635 | [0.890–7.796] | 0.080 | | | |
| Larerality 1 | | | | | | | | | | | | |
| Left | 1 | | | | | | 1 | | | | | |
| Right | 1.165 | [0.882–1.538] | 0.282 | | | | 1.246 | [0.885–1.753] | 0.208 | | | |
| Size 1 | 0.996 | [0.983–1.009] | 0.514 | | | | 0.995 | [0.980–1.011] | 0.537 | | | |
| ELN | | | **0.097** | | | 0.602 | | | **0.098** | | | 0.927 |
| 0–2 | 1 | | | 1 | | | 1 | | | 1 | | |
| 3–8 | 0.736 | [0.535–1.013] | **0.060** | 1.087 | [0.775–1.524] | 0.629 | 0.677 | [0.459–0.997] | **0.048** | 0.990 | [0.659–1.487] | 0.962 |
| >8 | 0.714 | [0.503–1.013] | **0.059** | 0.911 | [0.621–1.336] | 0.634 | 0.685 | [0.448–1.047] | **0.080** | 0.920 | [0.580–1.460] | 0.724 |
| Second tumor | | | | | | | | | | | | |
| Primmy site 2 | | | **<0.001** | | | **0.038** | | | **0.004** | | | 0.357 |
| Upper lobe | 1 | | | 1 | | | 1 | | | 1 | | |
| Middle lobe | 1.264 | [0.749–2.135] | 0.380 | 1.299 | [0.762–2.215] | 0.336 | 1.029 | [0.516–2.052] | 0.936 | 1.043 | [0.518–2.101] | 0.905 |
| Lower lobe | 0.918 | [0.689–1.223] | 0.558 | 0.837 | [0.622–1.126] | 0.240 | 0.919 | [0.649–1.302] | 0.635 | 0.887 | [0.620–1.269] | 0.511 |
| Overlapping lesion | 7.936 | [2.893–21.769] | <0.001 | 3.364 | [1.178–9.610] | **0.023** | 8.405 | [2.613–27.033] | <0.001 | 2.715 | [0.795–9.266] | 0.111 |

(Continued)

| Variable | Overall survival | | | | | | Cancer-specific survival | | | | | |
|---|---|---|---|---|---|---|---|---|---|---|---|---|
| | Univariate | | | Multivariate | | | Univariate | | | Multivariate | | |
| | HR | 95% CI | P | HR | 95% CI | P | HR | 95% CI | P | HR | 95% CI | P |
| Grade 2 | | | **0.072** | | | 0.096 | | | **0.032** | | | **0.048** |
| I | 1 | | | 1 | | | 1 | | | | | |
| II | 1.549 | [1.048–2.289] | **0.028** | 1.463 | [0.983–2.177] | 0.061 | 1.622 | [0.984–2.671] | **0.058** | 1.570 | [0.946–2.605] | 0.081 |
| III | 1.571 | [1.038–2.377] | **0.033** | 1.188 | [0.766–1.844] | 0.442 | 1.939 | [1.157–3.247] | **0.012** | 1.584 | [0.921–2.725] | 0.097 |
| IV | 2.552 | [0.993–6.560] | **0.052** | 2.710 | [0.982–7.479] | 0.054 | 3.591 | [1.224–10.534] | **0.020** | 4.811 | [1.516–15.267] | **0.008** |
| Laterality 2 | | | | | | | | | | | | |
| Left | 1 | | | | | | 1 | | | | | |
| Right | 0.876 | [0.670–1.146] | 0.334 | | | | 0.829 | [0.597–1.151] | 0.262 | | | |
| Size 2 | 1.018 | [1.005–1.032] | **0.008** | 1.031 | [1.015–1.047] | **<0.001** | 1.017 | [1.001–1.034] | **0.039** | 1.030 | [1.011–1.049] | **0.002** |
| ELN 2 | | | **0.001** | | | **<0.001** | | | **<0.001** | | | **<0.001** |
| 0 | 1 | | | 1 | | | 1 | | | 1 | | |
| 1–9 | 0.694 | [0.523–0.922] | **0.012** | 0.676 | [0.504–0.906] | **0.009** | 0.697 | [0.497–0.978] | **0.037** | 0.689 | [0.488–0.973] | **0.035** |
| >9 | 0.450 | [0.283–0.716] | **0.001** | 0.411 | [0.253–0.668] | **<0.001** | 0.264 | [0.132–0.530] | **<0.001** | 0.223 | [0.109–0.459] | **<0.001** |

**Notes:**
MDP-NSCLC, metachronous double primary non-small cell lung cancer; HR, hazard ratio; CI, confidence interval; ELN, examined lymph node.
The bold entries represent statistically significant.

to the SDP-NSCLC group. However, lesions have also been observed in the ipsilateral lung by other researchers (*Yu et al., 2013*). The grades of the lesions among all observed T1-2N0M0 patients were mostly well-middle differentiated, reflecting the observations made by *Arai et al. (2012)*.

Metachronous multiple primary cancers appear to have better survival rates than synchronous multiple primary cancers (*Baba et al., 2018*), so did lung cancer in our study. This may be due to the short interval between the occurrence of two tumors, which increases the tumor load. Our research indicated that age and sex were independent prognostic factors for both OS and CSS in the SDP-NSCLC group while grade max was related only to CSS, which is consistent with previous studies (*Shan et al., 2017*; *Yu et al., 2013*). The prognostic factors of OS and CSS were more likely associated with the second primary cancer and especially the tumor size, according to survival analysis of patients with MDP-NSCLC (*Gulack et al., 2015*). The reason for this result may be that the interval time between the two primary cancers in the MDP-NSCLC group is long and the first primary T1-2N0M0 lung cancer after radical surgery can be regarded as complete remission. Besides, our study also showed that the longer interval time was indicative of a better prognosis, which can be attributed to the second primary cancer being less invasive (*Aziz et al., 2002*). A recent meta-analysis on the location of the lesion on the ipsilateral or bilateral side revealed that location had no effect on the prognosis of DPLC patients, which was confirmed by our results (*Jiang et al., 2015*).

Combined with other research, it is widely accepted that the lymph node involvement is highly associated with the prognosis of multiple primary cancers except the factors we

have identified (*Loukeri et al., 2015*; *Voltolini et al., 2010*). We focused on the ELN count and analyzed its relationship with prognosis since T1-2N0M0 DP-NSCLC patients are free of lymph node metastases and are typically treated with surgery. Previous studies reported that the amount of ELN may be beneficial to survival in early-stage single primary NSCLC but there was no consensus on the optimal ELN count during curative lung cancer surgery. *Becker et al. (2018)* suggested that more than 16 ELNs may lead to improved survival outcomes, while *Osarogiagbon, Ogbata & Yu (2014)* concluded that the lowest mortality risk occurred in patients with 18–21 ELNs. Surgeries for SDP-NSCLC patients with two lesions were conducted simultaneously or within a short period of time so that lymph node dissection might overlap or influence each other. To overcome this, ELN plus was used and found to be an independent prognostic factor of both OS and CSS in the SDP-NSCLC group. We recommended a higher ELN count than previous studies of single primary lung cancer, which may be due to the fact that patients with double primary cancer have a heavier tumor load, worse prognosis and higher risk of lymph node metastases requiring more lymph nodes to be examined. Patients have a greater chance of survival with an ELN plus greater than 22. In MDP-NSCLC patients, a total of nine or more ELNs of the second primary cancer were associated with greater survival. These figures reflect a lower optimal ELN count than those obtained by previous researchers for single primary lung cancer. It is possibly that part of the lymph nodes that need to be examined during the radical surgery of the second primary NSCLC have been cleaned in the operation process of the first primary NSCLC.

The reason for the association between ELN count and survival is unknown, however, it is may be the result of more accurate staging. Previous studies indicated that not all patients with negative lymph nodes were free of lymph node metastasis. *Wang et al. (2015)* investigated 292 patients with IA peripheral lung cancer who underwent surgery and later discovered lymph node metastasis in 10% of those patients. *Osarogiagbon et al. (2011)* demonstrated that large proportions of pN0 patients were likely understaged. These studies indicate that the examination of a greater number of lymph nodes can maximize the probability that positive nodes are detected. Node-positive patients who were clinically understaged as T1-2N0M0 according to NSCLC guidelines would be less likely to receive adjuvant therapy after surgery, causing delayed treatment. ELN count is important in the assessment of the prognosis and risk stratification of early-stage lung cancer. This theory may be also applicable to early-stage DP-NSCLC.

There are several potential limitations in our study. First of all, it is limited by its retrospective nature for potential selection and time biases. Second, there is also no standardized ELN count, the count is assessed by pathologists based on their clinical experience. Third, some information is not provided in the SEER database including LN stations, LN sides, performance status, quality of life, driver gene status, additional chemotherapy and targeted therapy and comorbidities. Finally, tumor size and differentiation may impact the optimal number of ELN and thus affect survival (*Gulack et al., 2015*). Therefore, large prospective studies on the first-rank cutoff of the ELN count for early-stage DPLC classified by tumor size and differentiation should be conducted for further analysis.

## CONCLUSIONS

Our study indicated that ELN count is an independent prognostic factor for T1-2N0M0 double primary NSCLC patients after surgery. More than 22 ELNs were recommended for SDP-NSCLC patients, while more than nine lymph nodes should be examined during surgery on the second lesion for MDP-NSCLC patients.

### Funding

The authors received no funding for this work.

### Competing Interests

The authors declare that they have no competing interests.

### Author Contributions

- Kan Jiang conceived and designed the experiments, performed the experiments, analyzed the data, prepared figures and/or tables, authored or reviewed drafts of the paper, and approved the final draft.
- Xiaohui Zhi conceived and designed the experiments, performed the experiments, analyzed the data, prepared figures and/or tables, authored or reviewed drafts of the paper, and approved the final draft.
- Yue Shen analyzed the data, authored or reviewed drafts of the paper, and approved the final draft.
- Yuanyuan Ma performed the experiments, prepared figures and/or tables, and approved the final draft.
- Xinyu Su analyzed the data, authored or reviewed drafts of the paper, and approved the final draft.
- Liqing Zhou conceived and designed the experiments, authored or reviewed drafts of the paper, and approved the final draft.

### Data Availability

The raw data are available in the Supplemental Files.

### Supplemental Information

Supplemental information for this article can be found online at http://dx.doi.org/10.7717/peerj.8692#supplemental-information.

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
