# Peer review of "Impact of examined lymph node count on long-term survival of T1-2N0M0 double primary NSCLC patients after surgery: a SEER study"

_PeerJ, doi:10.7717/peerj.8692_

## Round 0.1 · original submission · Major Revisions

Manuscript entitled "Impact of examined lymph node count on long-term survival of T1-2N0M0 double primary NSCLC patients after surgery" which you submitted to PeerJ, has been reviewed. The reviewers have recommended publication pending major revisions. Therefore, I invite you to respond to the reviewers' comments at the bottom of this letter and revise your manuscript accordingly.

·

Basic reporting

1. In Figures 1 and 2, x-tile is best used only to determine the cut-off value. Patients should be regrouped according to this cut-off value and survival analysis should be conducted by new groups. You can put the picture of the x-tile results in the supplementary picture.
2. In lines 131-132, the authors say “no significant difference was found in sex and race between two groups.” No specific p value is given here. Even if there is no significant difference between variables, the p value is best listed.
3. In lines 137-138, the authors say “Among all lesions, the median ELN count went out to be 4 but differed significantly between two groups (P=0.018) possibly due to the diverse distribution.” Only objective facts need to be stated in the results section, and the reasons for the differences in results should be discussed in the discussion section.

Experimental design

1. Patients are included between 1988-2013; it is unclear what the average follow-up time was, what was the last date patient information was updated? How does the SEER provide an update on OS or CSS (usually once yearly?). This should be specified.
2. This is a clinical retrospective study, and given that the data are from a public database, the authors would be well advised to state that the study was exempted by the ethics review committee.
3. Since there are too many criteria for inclusion and exclusion of patients, it is better to provide a flow chart to show how many patients are included and excluded in each step.
4. As far as I know, the number of examined lymph nodes in SEER database has some unreasonable data. How do you deal with some obviously unreasonable data? This needs to be specified in the methods section. If data from other database or medical institution can be used for external verification, it will help to solve this problem.
5. The authors say “SDP-NSCLC group patients were those which second primary NSCLC was identified within 6 months after the diagnosis of first primary NSCLC.” How is the 6 months standard formulated? Why not 3 months, 4 months or 9 months? What is the effect of different time nodes on the prognosis of the two groups?

Validity of the findings

1. In the abstract section, the authors conclude that “These findings indicated that ELN count played a vital role in the long-term survival of T1-2N0M0 double primary NSCLC patients after surgery.” In my opinion, this is not accurate. ELN count does not play an important role in prognosis, but is maybe highly correlated with prognosis. There is no causal relationship between them, so it cannot be said that ELN count plays an important role in the prognosis.

Additional comments

Lymph node dissection is an important part of tumor surgery. The pathological examination report of lymph nodes is helpful for the postoperative pathological stage of patients, which is also helpful for the clinicians to judge the prognosis of patients and the formulation of follow-up treatment strategies. The authors want to explore the relationship between the number of lymphadenectomy and the prognosis of multiple primary tumor patients. This is good, but there are still some questions confusing me and they need to be addressed furtherly.

Reviewer 2 ·

Basic reporting

The paper needs to be check by a professional native-english speaker.

Experimental design

The design of the study is well presented into the methods; however, the results section is confusing and the author should better explain his finding adding more tables; otherwise, it is very difficult to read and understand the finding.

Validity of the findings

the result section is confusing and difficult to read; the author should better explain the finding eventualy adding more tables to make it more clear to the reader . Also the discussion is more a review of the literature than a discussion of the results of the study and it should be re written with more discussion of their finding compare with the literature.

Additional comments

The manuscript is about a very interesting topics; however there are few issues that the author should fix before considering it for publication. In particular the result section is confusing and difficult to read; the author should better explain the finding eventualy adding more tables to make it more clear to the reader . Also the discussion is more a review of the literature than a discussion of the results of the study and it should be re written with more discussion of their finding compare with the literature.
- This sentence into the methods line 97-100 should be re written because it is not clear: "On account of the difficulty in discriminating the first and second primary cancer in SDP-NSCLC, we brought each lesion rather than patient into the statistics and set up new variables, including grade max, size max and ELN plus, in the survival analysis of SDP-NSCLC group"
- The author should better explain what the X-tile analysis is.
- Into the results section the paragraph between line 149-154 is unreadable. Please, make it more clear to the reader.
- Into the discussion the author stated: "Then, we recommended that a sum of more than 22 lymph nodes should be examined during the surgery of two lesions for SDP-NSCLC patients while over 9 lymph nodes to be examined during the surgery of the second lesion for MDP-NSCLC patients, with the purpose of achieving better survival " How many per each side? Which stations?? Hilar? Mediastinal? Considering that this sentence is the more important one of the study (see the conclusion), it should be better explained and argumented into the discussion.

---

## Round 0.2 · accepted · Accept

Thanks for the revision of the manuscript, which can be now accepted.

·

Basic reporting

no comment

Experimental design

no comment

Validity of the findings

no comment

Additional comments

no comment